# Social Justice, Food Loss, and the Sustainable Development Goals in the Era of COVID-19

**Janet Fleetwood**

Department of Community Health & Prevention, Dornsife School of Public Health, Drexel University, Philadelphia, PA 19104, USA; janet.fleetwood@drexel.edu

**Abstract:** The United Nations' Sustainable Development Goals (SDGs) rest on a set of broadly accepted values within a human rights framework. The SDGs seek to improve human lives, improve the planet, and foster prosperity. This paper examines the human rights framework and the principles of social justice and shows that, while the SDGs do not specifically state that there is human right to food, the SDGs do envision a better, more just, world which rests upon the sufficiency of the global food supply, on environmental sustainability, and on food security for all. Then the paper examines the interrelationships between the SDGs, food access and waste, and human rights within a framework of social justice. Finally, it looks at the potential pandemic of hunger wrought by COVID-19, showing that COVID-19 serves as an example of a crisis that has raised unprecedented challenges to food loss and waste in the global food supply system and tests our commitment to the principles espoused by the SDGs.

**Keywords:** food loss; food waste; social justice; sustainability; COVID-19

## 1. Introduction

The Sustainable Development Goals (SDGs) set forth universal goals to enhance human lives, improve our planet, and foster prosperity. Addressing food loss and waste fosters better health while simultaneously improving our ecosystem by reducing the negative environmental impact of food waste. This paper examines the human rights framework used in the SDGs, considers social justice, and shows that, while the SDGs do not specify a human right to food, the SDGs describe a vision which rests upon the sufficiency and sustainability of the food supply. Moreover, common principles of social justice such as those developed by John Rawls support the need to address food insecurity and sustainability for our planet.

The literature on sustainability, food security, social justice, and food loss and waste is large and varied, ranging from technological assessments and proposed solutions, to economic analyses, to social and philosophical examinations of food as a human right. This paper examines the interrelationships between the SDGs, food access and waste, and human rights within a framework of social justice. Finally, this paper highlights how the impact of COVID-19 will test our commitment to social justice and to the SDGs as it impacts the global food supply chain, and food loss, as never before.

To begin, this paper briefly discusses some key concepts related to sustainability and food security. Then, it highlights the specific human rights stipulated in the SDGs, including the right to safe water and sanitation, and examines the implication and possible reasons for the SDGs omitting mention of a specific human right to food. Next, it shows that, despite the absence of a specific human right to food in the SDGs, a classic social justice approach supports the need to reduce food loss and waste to protect future opportunities for others—an issue of keen concern within the SDGs. The paper concludes by looking at the current global food crisis brought on by COVID-19. Our current situation—which includes both dramatically increased hunger and increased food waste—shines a spotlight on some of

the problems in our food system and on the need for swift changes, such as those proposed even prior to the pandemic by the Food and Agriculture Organization of the United Nations. The SDGs may not explicitly state that there is a human right to food, but the many other SDG goals implore us to take a more aggressive stance now toward food loss and waste if we are to have any hope of avoiding a humanitarian food crisis.

## 2. Key Concepts Related to Food Loss and Food Waste

The Food and Agriculture Organization of the United Nations (FAO) recognizes four pillars of food security, including physical *availability* of food; economic and physical *access* to food at the international, nation, and household level; food *utilization* or the sufficient energy and nutrient intake for good nutritional status; and *stability* of the previous three dimensions over time [1]. There are many definitions of food security. According to the FAO, food security exists when "all people, at all times, have physical and economic access to sufficient, safe and nutritious food to meet their dietary needs and food preferences for an active and healthy life" [2].

The global goal of ending hunger is daunting, given that, even prior to the COVID-19 pandemic, approximately 820 million people had a caloric intake insufficient to meet minimum energy requirements, and about 1.9 billion people struggled with or worried about access or affordability of a healthy diet [3]. Then, as now, poverty and malnutrition form a vicious cycle as each contributes to the other; being poor makes one unable to afford or access nutritious food, and lack of nutritious food makes one less likely to be able to get an education or work sufficiently to relieve oneself of poverty. In addition, non-state and state-based restrictions or violence results have resulted in internal and international displacements which have made ensuring a steady nutritious food supply extremely challenging—challenges which have been exacerbated during the COVID-19 pandemic [4].

Food loss and food waste solutions include both upstream and downstream interventions, such as waste prevention at the agricultural and production level, redesigning business processes and supply chains, and improving consumers' knowledge and behaviors related to food waste [5]. However, defining what counts as food loss or food waste, and then quantifying it, are notoriously difficult. Research about food loss and food waste is especially complicated by the various definitions that differentiate between food loss and food waste, and sometimes the terms are used interchangeably [6]. Further, methods of measuring loss and waste have historically been inconsistent or unreliable [7–12].

For the purpose of this discussion, the 2016 Food Loss and Waste Accounting and Reporting Standard (FLW Standard) provides guidance on measuring and reporting and describes what to measure and how to measure it. The FLW Standard provides a modular definition of food loss and waste, which can be customized relative to the materials and destinations of each organization's goals, while still being internationally consistent. As the FLW protocol authors explain, "an entity may choose whether it quantifies both food and inedible parts removed from the food supply chain, only food, or only associated inedible parts, as well as which destinations will be included within its scope." Examining the distinctions and the measurement methods is beyond the scope of this paper, but this paper can safely use the aggregate term used by the Food Loss and Waste Accounting and Reporting Standard, which considers "the weight of food and/or associated inedible parts removed from the food supply chain, which is referred to as 'FLW'" [13]. We will therefore consider food loss and food waste together as "FLW".

Addressing food security by addressing FLW requires analysis of the food chain from production to consumption. A pivotal analysis about sustainable development, published by the United Nations in 2015, seeks to cut per capita global food waste in half at the retail and consumer levels and reduce food loss along the entire production and supply chain [14]. Given that about one third of all food produced is lost or wasted each year, food loss and waste are reasonable places to start the process of eliminating hunger, food insecurity, and malnutrition [15]. Edible food is lost or wasted during production, handling and storage, processing and packaging, distribution and market, and consumption [16]. Such losses impede food security, while increasing environmental damage and economic costs.

There is general agreement that the world produces enough food for everyone, if only we could ensure that nutritious and culturally appropriate food is available when and where people need it, enable safe access to it, and guarantee the resources to obtain it. Yet reducing global FLW is a complex goal, replete with sociocultural, political, and ethical issues.

## 3. Sustainable Development Goals, Human Rights, and FLW

The Sustainable Development Goals build on the eight Millennium Development Goals, signed in September 2000, and rest on a human rights framework and the basic values of equality and nondiscrimination [17]. The 17 SDGs and 169 targets reflect an ambitious agenda which balances economic, social, and environmental elements of sustainable development. The vision resolves by 2030, "to end poverty and hunger everywhere; to combat inequalities within and among countries; to build peaceful, just and inclusive societies; to protect human rights and promote gender equality and the empowerment of women and girls; and to ensure the lasting protection of the planet and its natural resources" [18]. Achieving the SDGs will require efforts by governments, parliaments, the United Nations, and the private sector to collaborate in multistakeholder partnerships [19]. The efforts have been ongoing, yet the COVID-19 pandemic threatens our food system from production to consumption, in countries around the world.

However, first, what is meant by a "human right"? Is there a universal human right to food? In short, human rights are norms and freedoms that guarantee basic needs and protect all people from abuses. According to the United Nations in the Universal Declaration of Human Rights of 1948, human rights are "inherent to all human beings" and include the right to life, liberty, freedom from slavery or torture, freedom from arbitrary arrest or detention, freedom of opinion and expression, the right to work, the right to education, and others [20]. Human rights include both positive rights—the right to be provided with something, such as clean and safe water—and negative rights—the right not to be subjected to something, such as the right to be free from slavery and torture. Food is specifically mentioned in Article 25 of the Universal Declaration as part of the right to a standard of living "adequate for the health and well-being of himself and of his family" [21].

International human rights laws specify the obligations of governments to promote and protect human rights. The right to food has been discussed at several international summits since 1948. In 1996, at the Rome Declaration on World Food Security, all countries except Australia and the United States agreed that food is a basic human right [22]. The Food and Agriculture Organization of the United Nations has asserted that the human right to food is the right "to feed oneself in dignity" which must be fulfilled by "governments and non-state actors" and is not simply a right to charity [23]. The right is founded on the accountability of the government and nonstate actors to assess food insecurity, to create plans to address food insecurity, and to attempt to effectively implement those plans—not on the voluntary donations of private individuals who give charity to those who cannot afford sufficient, nutritious food.

The Sustainable Development Goals are guided by the United Nations' Universal Declaration of Human Rights, various international human rights treaties, and previous United Nations conferences and summits [18]. The SDGs specifically resolve to protect human rights and "realize the human rights of all" [18,24]. Human-rights-friendly countries generally support the human right to food, and the Universal Declaration of Human Rights recognizes the right to food as part of an adequate standard of living. Yet, the human rights that are explicitly recognized in the SDGs are limited to mentioning of access to safe water and sanitation, where it is stated, "A world where we affirm our commitments regarding the human right to safe drinking water and sanitation and where there is improved hygiene; and where food is sufficient, safe, affordable and nutritious" [18,25]. We must also take note of the connection between FLW and the right to adequate food, and the obligations or states in implementing rights, as the FAO has done in their 2018 discussion paper [26]. Notably, while food security is described as a "development priority" in the SDGs alongside poverty eradication, health, and education, food security and nutrition are not stated as specific human rights. In fact, that

"affordability" criterion in the vision exposes a gap between the duty to ensure sufficient safe and nutritious food and the less stringent obligation to assure that the food is affordable [18]. Is it simply up to the free market to determine who eats and who does not?

The SDGs list other rights aside from safe water, but they are not generally rights to other material goods. Instead, the SDGs refer to human rights and to "reproductive rights," "equal rights to economic resources," and "labor rights" [18]. The omission of food as a human right likely reflects the spirit of compromise necessitated by differences in a small number of highly influential countries' interpretation of the appropriate scope and limit of human rights. While all could agree on the right to safe water, not all could agree on the right to food, so it was not explicitly named as a human right in the SDGs.

## 4. FLW, Sustainability, and Social Justice

Nevertheless, food security, environmental impact, and food loss and waste are clearly priorities in the SDGs which discuss the need to "ensure access by all people . . . to safe, nutritious and sufficient food all year round" and "eradicate all forms of malnutrition" [18]. The clearest connection between food security and FLW is SDG 2, "End hunger, achieve food security and improved nutrition and promote sustainable agriculture," while #12, responsible consumption and production, can be interpreted to refer to managing food waste on the industrial and household level [18]. Moreover, the SDGs state, "We are committed to achieving sustainable development in its three dimensions—economic, social, and environmental—in a balanced and integrated manner" and the SDGs specifically refer to "environmental protection and the eradication of poverty and hunger" [18]. Finally, they seek to improve "global resource efficiency in consumption and production and endeavour to decouple economic growth from environmental degradation" [18]. Thus, the SDGs and environmental ethics are closely intertwined [27].

Our popular understanding of sustainable development dates back to the 1987 Brundtland Commission Report, *Our Common Future*, which describes "development that meets the needs of the present without compromising the ability of future generations to meet their own needs" [28]. Yet, as Berry points out, the idea of linking food security with sustainability originated in Asia in the early 1900s [29]. The idea of living well now while preserving the possibility for others to live well in the future is a key goal of sustainability and of sustainable development.

This approach to sustainability has much in common with the philosophic approach of John Rawls, an American philosopher who wrote the highly acclaimed book, *A Theory of Justice* [30]. Rawls is considered one of the most important political philosophers of the twentiethth century, and he received the United States National Humanities Medal in 1999. Distilled into its key components, Rawls sets forth a comprehensive theory of a just society. Based on Immanuel Kant's ethics and social contract theory, Rawls set out to find the principles that rational, free, and mutually disinterested people would choose to guide their society. To arrive at those principles he devised a hypothetical situation in which rational individuals, in what he called the "original position" behind a "veil of ignorance" pertaining to their talents, propensities, preferences, economic and social conditions, decide the rules for their imagined society. In short, nobody knows if they are rich or poor, well educated or uneducated, healthy or ill—struggling at the bottom of the social and economic heap or rejoicing at the tippy top. He argues, persuasively, that rational individuals behind the veil of ignorance who sought to create a just society would settle on two basic principles. First, people behind the veil of ignorance would decide that each person should have an equal liberty that is compatible with equivalent liberty for others; and second, social and economic inequality must (a) be attached to positions open to all with fair equality of opportunity and (b) must produce the greatest benefit to the least-advantaged members of society.

The essential elements of Rawls' core principles of justice are still used today in considering what is just and fair and can be used to examine the food security and sustainability issue. While a full philosophical analysis of Rawls' theory and his conception of intergenerational justice is beyond the scope of this paper, Rawls' theory clearly includes equality of opportunity and encompasses the need to bring forward the most vulnerable and the furthest behind. Rawls' approach reflects commitment

to distribute things equitably now, the need to improve the circumstances of those with the least, and the desire to preserve options for future generations—generations which will frame their own goals and conceptions of the good. From this, while we cannot deduce a "right to food," we can surmise that ensuring sufficient, nutritious food while reducing food waste and its attendant negative environmental impacts are reasonable social goals with the attendant societal obligations [31–33]. As the SDGs state, "As we embark on this collective journey, we pledge that no one will be left behind" and again, " . . . we will endeavor to reach the furthest behind first," and we will "have a particular focus on the poorest, most vulnerable and those furthest behind" [18].

A more recent analysis of social justice and sustainability which builds upon Rawls's work can be found in Curren and Metzger's 2017 book, *Living Well Now and in the Future: Why Sustainability Matters* [34]. Curren and Metzger argue that we need to focus on social coordination, with shared norms of cooperation and a common understanding of the problems for sustainability. They formulate a fundamental ethic of sustainability, which rests on concerns about social justice and on protecting our natural systems so that they can provide opportunities in the future which are comparable to what they provide to us now.

While the United Nation's SDGs do not specify a human right to food, the broader goals can be placed within concerns for social justice and sustainability. The United Nations' vision "seeks a world free of poverty, hunger, disease and want . . . where food is sufficient, safe, affordable and nutritious" and highlights the need "to reach the furthest behind first," and the need to create a world of sustainable resources [18]. Our obligation to protect the capacity of others in the future requires us to protect our environment, including protecting it from the negative impact of food loss and waste, and providing opportunity to all now and in the future.

## 5. The SDGs and the COVID-19 Food Crisis

The global pandemic of COVID-19 has sickened millions of people, killed hundreds of thousands, and caused unprecedented economic damage. In so doing, the pandemic has also illuminated many of the shortcomings of our global food production and distribution system, causing disruption to the food supply and food waste at a time when hunger is widespread and growing. SDG 3, "Ensure healthy lives and promote well-being for all," and SDG 8, "promote sustained inclusive and sustainable economic growth, full and productive employment and decent work for all," go hand in hand in the battle against food loss and waste during the era of COVID-19 [18]. This paper has briefly explored social justice and the SDGs which focus on alleviating hunger and on environmental sustainability related to food loss and waste, but the COVID-19 crisis puts our global food system under unprecedented strain. It will test the strength of the SDGs and the depth of our global commitment to social justice, sustainability, and food security.

The World Bank predicts that a hunger pandemic could swiftly follow the COVID-19 pandemic, doubling acute food insecurity by the end of the year, and that 40–60 million more people will be living in extreme poverty [35]. Not all of this is due to COVID-19 —droughts, pests including a locust infestation, and African Swine fever have contributed along with currency devaluations and depreciating commodities such as oil. However, COVID-19 brings extraordinary new challenges to hunger and FLW, and we will need new solutions [36].

In the higher-income countries, perishable foods, like meat and produce, have been most dramatically affected by COVID-19 due to their short shelf life and reliance upon workers for harvesting and processing. The SDGs explicitly mention protecting and empowering migrants and their contribution to sustainable development [18]. The SDGS call for "full respect for human rights" and "humane treatment" of migrants, refugees, and displaced persons irrespective of migration status [18]. These are the very migrants who are needed to harvest fields and for whom necessary travel restrictions to control the spread of COVID-19 stopped laborers' ability to follow the crops for work across Europe and the U.S. [37]. In addition, many farms were already working with labor shortages and reduced staff due to government restrictions on migration, making timely harvests

impossible. In short, a confluence of issues caused tens of millions of pounds of produce to rot in fields in the United States, Israel, and Europe. In the United States alone, farmers lost billions in revenue while hunger among jobless workers soared. In Europe, heavy reliance on complex supply chains, imported food, and just-in-time delivery all contributed to problems during COVID-19. If we are to rescue salvageable food, and avoid similar mistakes in the future, we must address our policies about migration.

Around the world, large-scale clients in the dormant hospitality industry, in closed schools, and in shuttered businesses no longer needed the food from fields and barns [38,39]. In the longer term, transportation of seeds and fertilizers was delayed, extending the global impact to the following year's crops [40]. Dairy farmers in the United States and United Kingdom dumped milk when demand in schools and restaurants dropped, and a spokesperson for the U.S. dairy industry called COVID-19's effect on the dairy industry "unprecedented in terms of its magnitude, its reach, and its complications" [41,42]. If we are to have the capacity to respond to disasters, we need to address our global supply chain issues.

To compound the negative impact of COVID-19 on perishable produce and dairy, large outbreaks of COVID-19 among workers in meat processing plants in the United States led to millions of chickens, pigs, and cattle being killed but not processed for food. Approximately 25% of U.S. meat processing plants were closed in two weeks during April 2020 due to massive COVID-19 outbreaks among thousands of meatpacking workers who work in close quarters with limited opportunity for social distancing and necessary hygiene. For example, at one point in the U.S. 700,000 pigs were euthanized weekly—some of which were composted to fertilizer while others were sent to landfills [43]. The workers and their families suffered, animals died unnecessarily, and FLW—and its attendant environment impact—soared [44–47]. Whether and how the SDGs will be invoked to deal with the working conditions of the meatpacking workers and the labor practices of the industry remains to be seen.

Meanwhile, ship cargos spoiled due to quarantines while restaurants had refrigerators of food that could not be sold and consumed. Volatility of retail sales made forecasting difficult, leading to shortages or waste at the retail level in middle- and high-income countries. Household hoarding in higher-income countries increased food waste due to overbuying and spoilage at the consumer level [48]. In short, COVID-19 exacerbated pre-existing problems in the global food system that previously could be minimized or even ignored. Yet we should not be surprised. The FAO's 2019 "State of Food Security and Nutrition in the World: Safeguarding against Economic Slowdowns and Downturns" focused on strategies for remediating food insecurity, hunger, and malnutrition in the context of sustainability during a crisis, examining the economic shocks felt in 33 countries affected by food crises in 2018 and 2019 and considering other potentially devastating events [49]. That document could not be more perspicacious or predictive of what was to come with the onslaught of COVID-19.

## 6. Conclusions

Problems of avoidable FLW and hunger have long been subjects for debate and scrutiny, but never has global waste from farm to fork been so visible and so urgent. As we consider social justice, sustainability, and the SDGs, we can see how the most fundamental values—of equality, of protection of opportunity, and of our shared humanity—underlie our global goals. Even without an explicitly stated human right to food in the SDGs, our notions of social justice and the need for environmental sustainability are widely shared. What remains to be seen, however, is how we will move through today's global crisis of hunger and sustainability caused by COVID-19, and whether our commitment to the SDGs and the values they uphold will be sufficient to guide our decision-making.

**Funding:** This work received no external funding.

**Conflicts of Interest:** The author declares no conflict of interest.

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
