# Peer review of "Social Justice, Food Loss, and the Sustainable Development Goals in the Era of COVID-19"

_sustainability, doi:10.3390/su12125027_

Round 1
Reviewer 1 Report
The manuscript was a pleasure to read and certainly worthy of publication. It is a wonderful conceptual piece that is quite thought provoking. It is, however, not for the faint of heart. Even the title reveals to a potential reader that a wide expanse of high inference material will be presented---social justice, food loss, sustainable development, and in the era of COVID-19 is a ton of stuff to take on. And a ton of conceptual/philosophical material is included--often with only a comment that something should be done about it the issue put forward in the section. Still, I admire the willingness on the part of the author to get right out there--even as the COVID-19 crisis continues. And that is the paper's strength--what should we be thinking about related to hunger, food insecurity, right now, in this moment and the moments to come.
I have only three suggestions: 1. Is it a paper, is it an article? Both terms are used in the abstract and the intro. Either one is fine, I think, but a consistent moniker I think would be helpful. And what kind of paper/article is it? A data driven one? A conceptual one? A philosophical review of the current moment? 2. In the intro the claim is made that the article "culls from the immense literature only those articles of greatest relevance,".....and I think it might be informative to the reader to know how this culling was accomplished--particularly since there is so much conceptual material/areas covered in the paper. How was "relevance" determined/decided? 3. Line 112 includes a pretty big claim, without much mention/reference to comparators--"The efforts have been ongoing, yet the COVID-19 pandemic threatens our global food system like no other threat in recent history."
And I apologize for my obsessive nature--line 85, safety should read safely, I think, and line 264 processing plans should read processing plants. Thanks.
Author Response
Thank you for your helpful critique of the paper. I appreciate the time and effort you spent to improve it and I hope that I have responded appropriately to your concerns. Here are my responses to your points:
Point 1: Is it a paper, is it an article? Both terms are used in the abstract and the intro. Either one is fine, I think, but a consistent moniker I think would be helpful. And what kind of paper/article is it? A data driven one? A conceptual one? A philosophical review of the current moment?
Response 1: Very good point. I agree. It is a paper, not a review article. Another reviewer had similar concerns. I removed the word "article" throughout and I now say, "This paper examines the interrelationships between the SDGs, food access and waste, and human rights within a framework of social justice." (lines 34-36).
Point 2: In the intro the claim is made that the article "culls from the immense literature only those articles of greatest relevance,".....and I think it might be informative to the reader to know how this culling was accomplished--particularly since there is so much conceptual material/areas covered in the paper. How was "relevance" determined/decided?
Response 2: You are right. It is not a review article. Although I used PubMed, Web of Science, and Google Scholar to find the articles which were winnowed into 73 citations, it is clear that this synthesis is far too broad to be considered a thorough review article of the SDGs, human rights, sustainability and food waste. I have replaced the word "article" with "paper" to eliminate any expectation by the reader that this is a review article.
Point 3: Line 112 includes a pretty big claim, without much mention/reference to comparators--"The efforts have been ongoing, yet the COVID-19 pandemic threatens our global food system like no other threat in recent history."
Response 3: Good point. I'm afraid I'm a bit myopic about COVID-19 as it has had a huge impact on food security and waste in the U.S. (where I live). We don't normally dump fresh produce and we haven't seen thousands of people lining up to get federal food boxes in many years. It is like nothing we have seen here in recent history. But, as you point out, the readership is global in nature and my claim was very broad. I have rewritten line 121 to say, "...yet the COVID-19 pandemic threatens our food system from production to consumption, in countries around the world."
I appreciate the "obsessive" comments! They are helpful. "Safety" has been replaced with "Safely" and "plans" is replaced with "plants." Thank you for reading so carefully. (Said by the person who always finds the spelling errors on the restaurant menu... My family is tired of hearing about them).
Reviewer 2 Report
This paper examines the concept of the right to food in the light of the SDGs and the Covid 19 pandemic. I have five reservations about the arguments advanced in the paper. The first three reservations focus on the right to food; the fourth reservation is about food loss and waste; and the fifth reservation is about Covid 19.
(1) The author claims she has selected papers of greatest relevance from the huge literature on the SDGs and human rights:
‘This article culls from the immense literature only those articles of greatest relevance to the current complex issue of the SDGs, human rights, food waste, sustainability, and social justice’ (lines 36-38)
But there are only three academic works in the References which explicitly cover human rights issues – Chilton and Rose; Rawls; and Caney.
(2) The paper claims the SDGs stipulate some specific human rights – such as the right to water – but not the right to food:
‘it highlights the specific human rights stipulated in the SDGs, including the right to safe water and sanitation, and examines the implication and possible reasons for the SDGs omitting mention of a specific human right to food’ (lines 42-44)
But the right to water is not explicitly stipulated in the SDG on water: SDG 6 stipulates the need to “Ensure availability and sustainable management of water and sanitation for all”. If the author claims this stipulation amounts to a declaration of the right to water, the stipulation in SDG 2 to "End hunger, achieve food security and improved nutrition, and promote sustainable agriculture” could be interpreted as a declaration of the right to food.
(3) The paper claims that despite the lack of a declaration of the right to food in the SDGs, social justice arguments justify such a declaration:
‘it shows that, despite the absence of a specific human right to food in the SDGs, a classic social justice approach supports the need to reduce food loss and waste to protect future opportunities for others’ (lines 44-46)
But the author fails to supply such a justification. She ignores the vast literature on human rights (such Rights by Peter Jones (1994)), and instead refers to Rawls A Theory of Justice, without demonstrating how that theory supports a right to food. Is she contending that the right to food is one of the basic rights, or an implication of the difference principle, that both arise out of Rawls’ original position scenario? The following passage suggests the author believes the right to food derives from Rawls’ difference principle:
‘Rawls’ approach reflects commitment to distribute things equitably now, the need to improve the circumstances of those with the least, and the desire to preserve options for future generations – generations which will frame their own goals and conceptions of the good. From this, we can surmise that ensuring sufficient, nutritious food while reducing food waste and its attendant negative environmental impacts are reasonable social goals with the attendant societal obligations’ (lines 200-205).
But even if we accept that Rawls’ difference principle justifies ensuring a ‘reasonable social goal’ that people have food, this is not the same as conferring upon people a right to food. The author criticises SDG 6 for failing to acknowledge a right to food, yet SDG 6 stipulates the need to ‘End hunger, achieve food security and improved nutrition’. If SDG 6’s stipulation to end hunger and achieve food security and adequate nutrition falls short of declaring a right to food, how does Rawls’ difference principle (maximisation of the position of the least advantaged group in society) endorse a right to food?
The author’s other attempts to justify a right to food are no more convincing. She alludes to two books which focus on sustainability – the Brundtland Report Our Common Future (1987) and Curren and Metzger’s 2017 book, Living Well Now and in the Future: Why Sustainability Matters. But she does not explain how either of these books provides a justification for the right to food.
(4) I am unsure how the section on food loss and waste fits into the paper. Is the author saying that the right to food could easily be met if steps were taken to reduce food loss and waste? If so, does she imply that an explicit affirmation of the right to food in SDG 6 would generate more effective steps to end food loss and waste? If so, what evidence does she have to support such a claim?
(5) It is unclear to me what the link is between the discussion of the right to food and the discussion of Covid 19. Is the author saying that Covid 19 has sharply increased the amount of hunger in the world, and therefore it is even more important for us to affirm the right to food in order to meet this chronic emergency? If so, can she explain why a greater emphasis on the right to food is likely to persuade states to devote more effort to food aid? The opposite reaction might be expected from President Trump, since, as Vivero Pol and Shuftan (2016) point out, the USA has vehemently opposed the notion of a right to food.
My view of this paper is that the author needs to address the above five reservations satisfactorily before it can be considered for publication.
Author Response
Thank you for your helpful critique of the paper. I appreciate the time and effort you spent to make useful, insightful and cogent remarks. Here are my responses to your points:
Point 1: The author claims she has selected papers of greatest relevance from the huge literature on the SDGs and human rights:
‘This article culls from the immense literature only those articles of greatest relevance to the current complex issue of the SDGs, human rights, food waste, sustainability, and social justice’(lines 36-38)
But there are only three academic works in the References which explicitly cover human rights issues – Chilton and Rose; Rawls; and Caney.
Response 1: I agree. Another reviewer had similar concerns. I used PubMed, Web of Science, and Google Scholar to find the articles which were winnowed into 73 citations, but it is clear that this synthesis is far too broad for be considered a thorough review article of the SDGs, human rights, sustainability and food waste. I have removed the statement you noted and replaced it with, "This paper examines the interrelationships between the SDGs, food access and waste, and human rights within a framework of social justice." (lines 34-36). The works cited on human rights (Chilton, Rawls, Caney, the UN Declaration, Vivero) are among the best work, but of course there are many others that would have been referenced in a comprehensive review. I have also replaced the word "article" with "paper" to eliminate any expectation by the reader that this is a review article.
Point 2: The paper claims the SDGs stipulate some specific human rights – such as the right to water – but not the right to food:
‘it highlights the specific human rights stipulated in the SDGs, including the right to safe water and sanitation, and examines the implication and possible reasons for the SDGs omitting mention of a specific human right to food’ (lines 42-44)
But the right to water is not explicitly stipulated in the SDG on water: SDG 6 stipulates the need to “Ensure availability and sustainable management of water and sanitation for all”. If the author claims this stipulation amounts to a declaration of the right to water, the stipulation in SDG 2 to "End hunger, achieve food security and improved nutrition, and promote sustainable agriculture” could be interpreted as a declaration of the right to food.
Response 2: I would agree if SDG 6 were the only place that water is discussed, but it isn't. The SDGs commit to a right to safe drinking water in "Our Vision, section 7" in which it states "...A world where we reaffirm our commitments regarding the human right to safe drinking water and sanitation and where there is improved hygiene; and where food is sufficient, safe, affordable and nutritious." This key phrase affirms a right to safe drinking water. In contrast, it does not say there is a right to food anyplace - even in the Vision; only that food be "sufficient, safe, affordable and nutritious." (Italics mine). "Affordable" refers to the capacity to buy food, not to food as a human right like safe drinking water. (Please see paper line 148-150 and citations 27,28).
Point 3: The paper claims that despite the lack of a declaration of the right to food in the SDGs, social justice arguments justify such a declaration:
‘it shows that, despite the absence of a specific human right to food in the SDGs, a classic social justice approach supports the need to reduce food loss and waste to protect future opportunities for others’ (lines 44-46)
But the author fails to supply such a justification. She ignores the vast literature on human rights (such Rights by Peter Jones (1994)), and instead refers to Rawls A Theory of Justice, without demonstrating how that theory supports a right to food. Is she contending that the right to food is one of the basic rights, or an implication of the difference principle, that both arise out of Rawls’ original position scenario? The following passage suggests the author believes the right to food derives from Rawls’ difference principle:
‘Rawls’ approach reflects commitment to distribute things equitably now, the need to improve the circumstances of those with the least, and the desire to preserve options for future generations – generations which will frame their own goals and conceptions of the good. From this, we can surmise that ensuring sufficient, nutritious food while reducing food waste and its attendant negative environmental impacts are reasonable social goals with the attendant societal obligations’ (lines 200-205).
But even if we accept that Rawls’ difference principle justifies ensuring a ‘reasonable social goal’ that people have food, this is not the same as conferring upon people a right to food. The author criticises SDG 6 for failing to acknowledge a right to food, yet SDG 6 stipulates the need to ‘End hunger, achieve food security and improved nutrition’. If SDG 6’s stipulation to end hunger and achieve food security and adequate nutrition falls short of declaring a right to food, how does Rawls’ difference principle (maximisation of the position of the least advantaged group in society) endorse a right to food?
The author’s other attempts to justify a right to food are no more convincing. She alludes to two books which focus on sustainability – the Brundtland Report Our Common Future (1987) and Curren and Metzger’s 2017 book, Living Well Now and in the Future: Why Sustainability Matters. But she does not explain how either of these books provides a justification for the right to food.
Response 3: Thank you for pointing out that my argument does not necessarily entail a human right to food. I have made my point clearer and hopefully avoided further confusion.
I agree - there is a huge literature on human rights and this paper does not argue for or against the human right to food. Making the argument that there is a right to food would require a different kind of paper, in a different journal. Were I to write that paper, I would absolutely include Jones and many others. However, this paper is written for a different audience with a different purpose, so I have relied upon the classic, wide-respected work of John Rawls to broadly shape the discussion.
I agree with you that "even if we accept that Rawls’ difference principle justifies ensuring a ‘reasonable social goal’ that people have food, this is not the same as conferring upon people a right to food." Hence I have not made that claim.The paper asserts that the SDGs do not affirm a human right to food, but that concerns for social justice are sufficient to demonstrate a societal obligation to ensure sufficient and safe food for everyone despite it not being specified as a right. I do not attempt to justify a right to food in this paper, as I hope I've now clarified. (Line 208).
The Brundtland Commission Report is cited for historical background to show that the popular idea of sustainability dates back to 1987, not to justify a right to food. I also cite Berry, who correctly pointed out that food security and sustainability were already linked in the early 1900's in Asia.
Finally, I cite Curren and Metzger's 2017 book because that book extends Rawls' social justice work to the food and sustainability conversation. As they state on page 72, "The work that sets this in motion and has dominated discussion is John Rawls's 1971 classic, A Theory of Justice, and his subsequent refinements and extensions... Together these works offer an impressively developed and defended vision of a just society... We will present Rawls's theory because it has dominated the landscape of political philosophy and because it is the most relevant point of departure for our own."(Italics mine). Curren's work really needs to be cited in this context.
Point 4: I am unsure how the section on food loss and waste fits into the paper. Is the author saying that the right to food could easily be met if steps were taken to reduce food loss and waste? If so, does she imply that an explicit affirmation of the right to food in SDG 6 would generate more effective steps to end food loss and waste? If so, what evidence does she have to support such a claim?
Response 4: I'm not relying on the right to food, or tying the right to food into reducing food loss and waste. Nor do I mean to imply that if only the SDGs would affirm the right to food, it would solve the food loss and waste problem. Sadly, that isn't the case. Instead, we need to recognize that, even without a "right to food," our ethical concerns about social justice should compel us to address global hunger.
Addressing global food loss and food waste is a necessary step that is even more pressing now given the impact on food loss and waste of COVID 19. As I say at line 294, "Even without an explicitly stated human right to food in the SDGs, our notions of social justice and the need for environmental sustainability are widely shared."
Point 5: It is unclear to me what the link is between the discussion of the right to food and the discussion of Covid 19. Is the author saying that Covid 19 has sharply increased the amount of hunger in the world, and therefore it is even more important for us to affirm the right to food in order to meet this chronic emergency? If so, can she explain why a greater emphasis on the right to food is likely to persuade states to devote more effort to food aid? The opposite reaction might be expected from President Trump, since, as Vivero Pol and Shuftan (2016) point out, the USA has vehemently opposed the notion of a right to food.
Response 5: COVID 19 has caused an unprecedented impact on the global food system and some believe it is likely to cause a hunger pandemic. I have not argued for a right to food, but I have argued that a classic interpretation of social justice would demand us to ensure sufficient food for all even without a specific human right. I needed to make this connection to convince those who do not support a human right to food (i.e. the U.S. and Australia) that, even if the SDGs don't assert a human right to food, there is still an obligation based on concerns for social justice. Addressing food waste - especially in light of the dramatic examples of industrial-level food waste caused by the pandemic - will be an essential element of ensuring food access.
I note in the paper that the US has opposed the "right to food" (line 135, cite 22) but argue that we do not need to assert such a right to create a socially just response to the food waste/hunger crisis.
And, yes, we can expect something terrible from Trump now and in the future. It's hard to predict what awful thing Trump will say or do next.